# Architecture and Data Knowledge of the Regional Data Center for Intelligent Agriculture

**Emil Doychev [1], Atanas Terziyski [2] , Stoyan Tenev [2], Asya Stoyanova-Doycheva [1,3,*] , Vanya Ivanova [1] and Pepa Atanasova [1]**

1   Department of Computer Systems, University of Plovdiv, 4000 Plovdiv, Bulgaria;
    e.doychev@uni-plovdiv.bg (E.D.); vantod@uni-plovdiv.bg (V.I.); pepa.atanasova@customs.bg (P.A.)
2   Department of Chemistry, University of Plovdiv, 4000 Plovdiv, Bulgaria;
    atanas@uni-plovdiv.net (A.T.); sttenev@uni-plovdiv.bg (S.T.)
3   Institute of Information and Communication Technologies, Bulgarian Academy of Sciences,
    1113 Sofia, Bulgaria
*   Correspondence: astoyanova@uni-plovdiv.bg; Tel.: +359-88-8668617

**Abstract:** The main task of the National Research Program "Smart crop production", supported by the Ministry of Education and Science of Bulgaria and approved by the Council of Ministers, is the development of a regional data center to facilitate the work of farmers. The regional data center is part of the implementation of a smart crop production environment called ZEMEL which provides personal assistants supporting the work of farmers. The environment provides intelligent services for crop analysis and prevention and assists farmers in performing basic tasks related to crop production. The objective of the proposed article is to present the implementation of the architecture, infrastructure, and data architecture of a regional data center in the Plovdiv region. In order to clearly present the results of this work, which are the architectural and physical implementations of a regional data center and the storage of dynamic data and background knowledge, a methodology consisting of several steps is followed: the system infrastructure of the data center and the data architecture are discussed; one of the local pieces of infrastructure, implemented in the Institute of Plant Genetic Resources (IPGR) in the town of Sadovo in the Plovdiv region, is presented in detail, including the different types of sensors and their connection to the data center in wheat cultivation; the data repositories are discussed where dynamic data and background knowledge are stored. The paper pays special attention to background knowledge developed as ontologies for winter wheat cultivation. The results are summarized by drawing some conclusions and recommendations for the design of the local infrastructure of the center and the stored data to improve its performance.

**Keywords:** regional data center; intelligent agriculture; ontologies



## 1. Introduction

Agriculture is an area of major importance for Bulgaria. Opportunities are being searched for the efficient use of fresh water, optimum use of chemicals and pesticides, and the maintenance and increase in yields in changing climatic conditions. The national strategy for the development of AI by 2030 gives a significant place to smart agriculture. In addition, a national research program called "Smart crop production" has been adopted as part of this strategy. One of the tasks of the program is to develop a pilot vertical IoT infrastructure for smart agriculture in the Plovdiv region, collecting data from this infrastructure in a regional data center, from where it can be used for analysis and prevention. The choice of Plovdiv is not a coincidence—Plovdiv is located in the Thracian Valley and is one of the most fertile regions in Bulgaria.

In conjunction with the above-mentioned national program, an environment for smart crop production called ZEMELA [1] is being developed. It has been created as a cyber-physical social space and is a reference architecture of the virtual physical space (ViPS) [2,3]

developed by a team of the DeLC Laboratory at the Faculty of Mathematics and Informatics of the University of Plovdiv.

The aim of the platform is to serve farmers when growing different agricultural crops. One of its main components is personal assistants [4]. The architecture of the personal assistants is built as a multi-agent system consisting of one interface agent and multiple operational agents. Typically, the actions of the interface agent are reminders, warnings, advice, etc. To fulfill their objectives, the personal assistants use information that is collected from crop fields and background knowledge about the development and characteristics of the monitored crops.

Another key part of the ZEMELA platform is the event engine (EE). It has the main task of monitoring the occurrence of different crop events and informing the personal assistants of their appearance. For this purpose, the EE uses knowledge about the physiological development of plants and knowledge about the environment of agricultural crops [5]. This information is stored in the ADC (Agriculture Data Center), which is another core component of the ZEMELA platform [6].

The implementation of the ZEMELA platform includes the development of a number of regional data centers in different areas of the country to collect data on crop cultivation, development, and environment. Based on this data, analyses and prevention are conducted for each crop in each of the districts. One of the regional data centers is in the Plovdiv region, the implementation of which is presented in this paper. The building of this center includes the development of the architecture, the infrastructure of the center, and data architecture. The detailed realization of one of the local pieces of infrastructure, implemented in the Institute of Plant Genetic Resources in the town of Sadovo in the Plovdiv region is presented, including the different types of sensors and types of data collected in the center when growing winter wheat. The architecture of the regional data center is also discussed as well as the data repositories where dynamic and background knowledge is stored. Dynamic knowledge includes the data that are collected based on the different IoT nodes deployed in the crop fields. Their storage is in relational and non-relational databases. Background knowledge is persistent knowledge about crop cultivation stored in ontologies. In this paper, special attention is paid to background knowledge and ontologies developed for winter wheat cultivation.

The proposed architecture of the regional data center and the architecture of the data stored in it are compatible with the development of the ZEMELA smart agriculture platform. The data center is an important part of the architecture of this platform and contributes to its transformation into a cyber-physical social space by storing and processing data obtained from different sensors and devices in a real environment. The regional data center also provides personal assistants in ZEMELA with background domain knowledge, developed in the form of ontologies.

The main advantage of the proposed architecture of the regional data center is that it offers storage not only of raw data from real environments, but also data that are used as background knowledge in the field, in the form of ontologies, used by the personal assistants of the ZEMELA platform. Another benefit is the project to develop cheaper sensors whose data can be matched against reference statistics. This would reduce costs for farmers who wish to use the platform. The processing of the sensor raw data is performed at a low level by developed software adapters. They provide the processed data in a format that is usable by the intelligent components.

## 2. Methods

The methods to create the regional data center in the Plovdiv region are divided into four main phases. The first phase is the building of a conceptual infrastructure of the data center for smart agriculture and a data architecture concept, which are presented in Section 3 of the article. The regional data center in Plovdiv collects information from various pieces of local data infrastructure in the area. The construction of local infrastructure for data collection is the second phase of the construction of the regional center. As an example,

the local infrastructure built in the Institute of Plant Genetic Resources in the town of Sadovo is presented, which was created to monitor the cultivation of winter wheat. The architecture at the local level is given in detail in Section 4 of the article. The third phase is the development of the architecture of the regional data center in Plovdiv, which includes two levels. The first one is for communication with the local layers to receive and store stream data coming from the sensor networks. The second is related to the provision of different services to farmers using different approaches to model agricultural scenarios. Section 5 of the article presents the architecture of the first level (the second level is not the goal of this article). The final, fourth phase of the implementation of the regional data center is the representation of the stored data and knowledge. The data stored in the regional data center can be divided into dynamic data and background knowledge. The dynamic data come from the various sensor networks, while the background knowledge is described as ontologies and represents domain knowledge—for crop development, crop and animal husbandry, etc. Part of this knowledge necessary for growing winter wheat is presented in Section 6 of the article.

## 3. Related Works

This section of the paper provides a literature review of studies in two main parts. The first part focuses on research by teams that have built IoT infrastructure for smart agriculture. The second part looks at studies that have combined IoT infrastructure with artificial intelligence. These studies were selected because the regional data center in the Plovdiv region focuses on developing IoT infrastructure for smart agriculture, and it is also part of the ZEMELA smart agriculture platform which uses intelligent components.

In recent years, numerous pieces of research and practical developments have been made to create IoT vertical infrastructures for smart agriculture. Various publications address more global issues of building IoT vertical infrastructure for smart agriculture and they offer more private solutions. The Internet of Things integrates several existing technologies, such as wireless sensor networks, radio-frequency identification, cloud computing, middleware systems, and end-user applications. In [7], several advantages and challenges of IoT are identified. The authors present an IoT ecosystem and how the combination of IoT and DA enables smart agriculture. In addition, future trends and opportunities are presented which are categorized into technological innovations, application scenarios, business, and marketability. In [8], a comprehensive review of emerging technologies for Internet of Things (IoT)-based smart agriculture is introduced. Unmanned aerial vehicles, wireless technologies, open-source IoT platforms, software-defined networking (SDN), network function virtualization (NFV) technologies, cloud/fog computing, and middleware platforms are referred to as emergent technologies for the agricultural IoT. A classification of IoT applications for smart agriculture is also provided, including smart monitoring, smart water management, agrochemical applications, disease management, smart harvesting, supply chain management, and smart agricultural practices.

An essential feature of modern agriculture is the integration of artificial intelligence with Internet of Things systems. In [9], an intelligent IoT system for smart agriculture is presented based on the concept of front–rear-end separation and the framework of the Model-View-ViewModel. With the help of this system, it is possible to handle complex business logic and integrate intelligent components relatively easily. The system consists of a remote data service platform, data collection terminals, and wireless data transmission using narrow-band Internet of Things (NB-IoT) modules. An algorithm for deep-learning-based plant disease and pest detection is also implemented in the system. Furthermore, the system has a convenient expansion interface and can be used as a basic development platform for various agricultural IoT applications, such as a soil environmental monitoring system or an intelligent disease and pest monitoring system. In [10], a novel, scalable and private geo-distance evaluation system is presented. The system provides geographic-based services by computing the distances between sensors and farms privately. The key idea is to perform efficient distance measurement and distance comparison on encrypted locations

over a sphere, by leveraging a homomorphic cryptosystem. Through extensive experiments with real datasets, the authors show that the system achieves private distance evaluation on a large network of farms. Furthermore, the application of mobile devices in agriculture IoT applications is demonstrated. In [11], the problem of the efficient path planning of robotic swarms is addressed, formulating it as a specific type of vehicle routing problem. Various state-of-the-art algorithms are employed to solve this problem in order to decide on the best approach for different agricultural topologies, tasks, and the number of robots available. An end-to-end system is proposed and evaluated, using the Internet/Web as an infrastructure and communication medium, taking GPS input data from map providers, identifying and applying the most suitable algorithm for the specific landscape and task, and finally producing GPS coordinates as routes for the robots to follow. Recommendations for further improvements are discussed such as exploring the factors that influence the willingness of users of smart agricultural Internet of Things to conduct information security measures; help the government, agricultural organizations, or related organizations to take effective measures; provide targeted information security education and training; and help to train users. The awareness of information security precautions promotes a willingness to adopt information security behaviors among users of the smart agricultural Internet of Things.

Many software developments offer a combination of IoT, ontologies, and services. For example, in [12], the authors describe an ontology-based architecture that can be used for precision agriculture applications. There are different structures that work together; these structures have their own sensors and triggers and can function according to their role in the field. In [13], the authors consider a scalable service-oriented architecture (ONTAgri) based on several groups of concepts. The two main groups are domain ontology and system ontology. Domain is further divided into services and basic parts of agriculture, e.g., irrigation, fertilization, etc. In [14], the authors propose the AgriOnt framework that can be used for smart agriculture. It has four domains: a geographic ontology, a business subdomain, an IoT-based subdomain, and an agriculture-based subdomain.

The considered developments for smart agriculture that combine IoT and artificial intelligence are mainly based on data collected from IoT nodes deployed in monitored sites and services, or from IoT nodes, ontologies, and services that provide different functionalities in the field of agriculture. Inferences are made using various intelligent machine-learning algorithms. Although it is difficult to form a clear picture of the implementation of these systems from the exposition in the reference works, it can be inferred that our proposed architecture of a regional data center stores crop development data obtained not only from IoT sensors but also data that are specific and basic to the area in which the crops are grown. This knowledge is implemented as ontologies that can be reused for other regions in Bulgaria. The data are processed at a low level in the data center architecture, to be made available in a form convenient for processing by intelligent agents in the ZEMELA platform. These intelligent agents make their inferences and conclusions about crop development based on a combination of data from the IoT nodes and background knowledge about the field in the ontologies.

The realization of sensor networks requires large investments from farmers. In our research, we have made a proposal to use cheaper sensors along with high-quality ones to serve as reference points as the synchronization of the data accuracy is completed automatically before being written to the data stores. Processing and adjustment of the raw data are conducted at a low level in the architecture, using software adapters developed for different sensor types or data packet types. The architecture of the regional data center developed in this way enables its integration into independent workflows, analyzing data in real time like the intelligent components or services.

## 4. System Infrastructure and Data Architecture Overview

### 4.1. System Infrastructure

The infrastructure is built on three levels: local (edge), regional, and national.

Local level—the task at the local level is to deliver measurable data from the physical locations of interest. Currently, three sensor networks have been built at the local level, which are constantly expanding and improving. The first sensor network is built in the Institute for Plant Genetic Resources, located in the town of Sadovo (Plovdiv region). The second sensor network is deployed in the training pasture of the Agricultural University of Plovdiv. The third sensor network is placed in a tomato greenhouse located on the territory of the Maritsa Vegetable Crops Research Institute of Plovdiv, which is a national research center for scientific and scientifically applied activities as well as extension services in the field of vegetable crops, potato breeding, and technologies for growing vegetable crops.

Regional Data Center—the regional data center operates as a repository for data that are continuously arriving from the local level (the sensor networks) and for the specialized knowledge of agriculture. The center is equipped with a powerful server configuration and is located on the territory of the University of Plovdiv "Paisii Hilendarski". Furthermore, using different modeling approaches, the center prepares different types of analyses to support agricultural systems. For the regional center, the receiving and storage of data from the local level are mainly considered in this paper. The modeling and analyzing are performed in a platform (known as ZEMELA [1]) also deployed in the regional data center.

National Data Center—the National Data Center is being built at the Institute of Information and Communication Technologies of the Bulgarian Academy of Sciences. Its task is to summarize the information received from the regional data centers and to prepare global trend analyses for the agricultural sector.

In this article, the local and regional levels of this infrastructure are presented in more detail. To demonstrate the local level, the first sensor network is considered. The first sensor network is built in the Institute for Plant Genetic Resources. For the regional data center, the part responsible for receiving and storing the data obtained from the sensor level is discussed. Data and knowledge in the regional data center are presented in an ontology view. The ZEMELA platform is not presented in this article.

### 4.2. Data Architecture

The unification and combination of data obtained in different spheres, time, and production sections of agriculture will allow the creation of digital models of natural resources for individual farms, regions, and for the country in general. This approach to the creation of digital models can be called the decentralized method. It differs from the centralized method, in which information bases are formed on a global level with the help of space or other large-scale technologies. Precisely, the proposed method of the upward approach, which is the suggested bottom-up digitalization approach, will enable the construction of truly interactive, intelligent technologies based on big data, AI, ontologies, etc.

After building the data architecture (Figure 1), the focus will shift to the transformation of data into information, controlled by intelligent systems in real time.

PostgreSQL (v 14.5) [15,16] and the TimescaleDB [17] extension (on UBUNTU OS, version 22.04 LTS) are installed in the regional center. The collection of data from the sensors (a lot of data from a large number of sensors) for further analyses and the specificity of this data, namely time-series data, led us to the conclusion to use TimescaleDB. It is implemented as an extension of PostgreSQL and combines the convenience of a relational database with the speed of NoSQL databases.

One of the most important issues in organizing the IT infrastructure of the center is ensuring sustainable work. In order to increase stability, we are planning to use technologies of hardware and the information reservation of the servers and the processed information. We intend to use replication for the data in PostgreSQL—Streaming Replication (Figure 2) with one primary and one secondary server that are geographically distant from each other.

We will use asynchronous replication in order not to slow down processing on the primary server, as the data from the sensors is time-series and it comes with a rather high intensity. We will employ replication for two main purposes:

- Increasing failure resistance—if the primary/master fails, we will be able to connect to the replica/secondary.
- Raising performance—for analytical calculations, we will target the secondary.

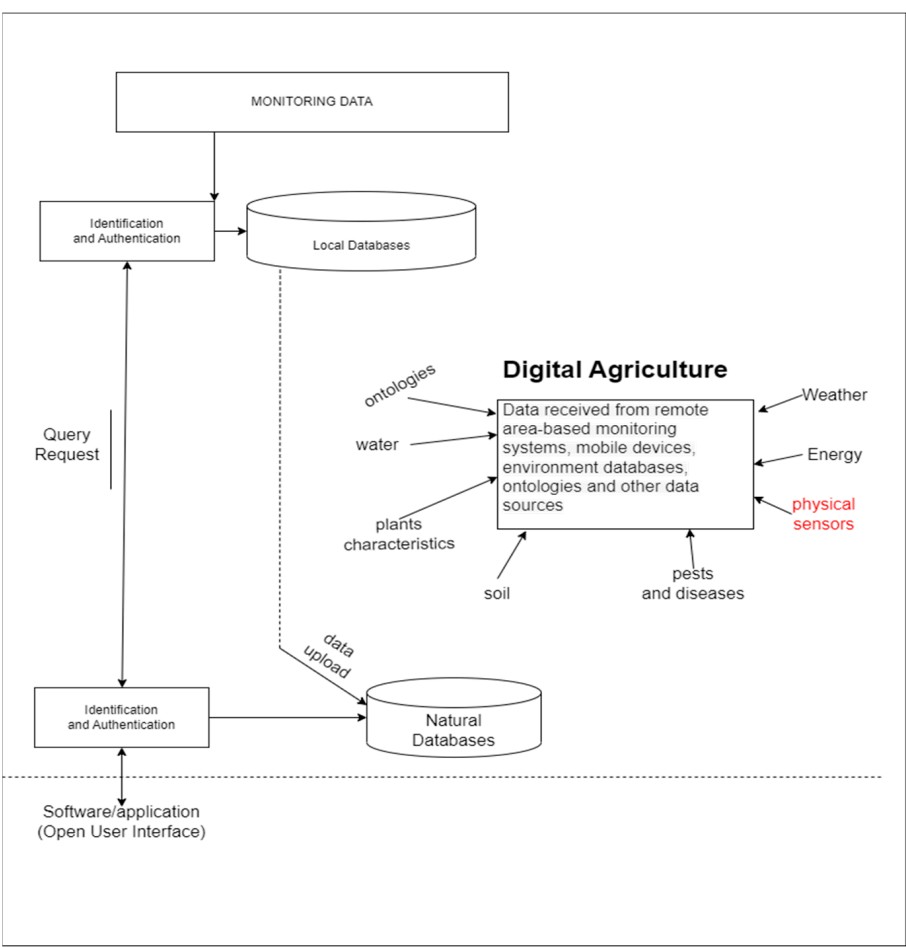

**Figure 1.** Data Architecture.

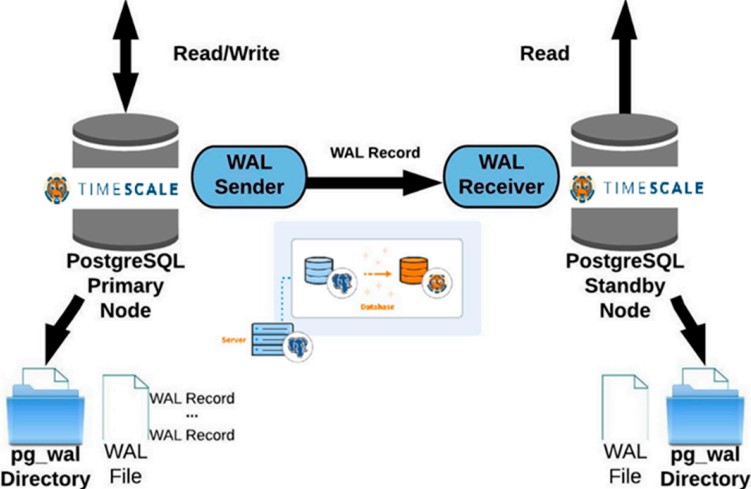

**Figure 2.** PostgreSQL and TimescaleDB—Streaming Replication.

Mongo DB 6.0 [18] is also installed in the regional center.

With reference to MongoDB, we provide replication—a replica set in MongoDB (Figure 3) (a group of processes that have passed through MongoDB that keep up the same data set). Replica sets provide redundancy and high availability, and they are the foundation for all production deployments. Replication provides redundancy and increases data availability. With multiple copies of data on different database servers, a level of replication provides failure resistance against the loss of a single database server. In our case, replication can provide increased read capacity because clients can send read operations to different servers (primary or secondary). Keeping a copy of data in different data centers can increase data locality and availability for distributed/decentralized applications.

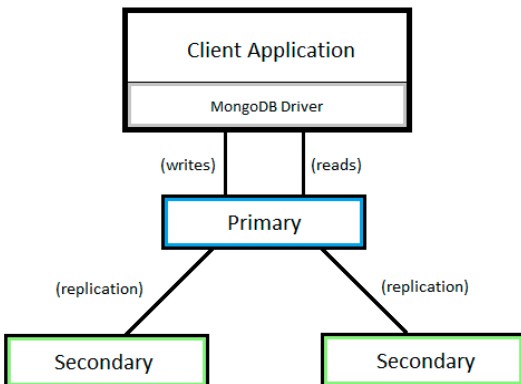

**Figure 3.** Replica set in MongoDB.

If the sensor data grows significantly over time, we are planning to use sharding for the MongoDB data. Horizontal scaling allows almost unlimited scaling for the processing of large amounts of data and intensive workloads.

Last but not least, we should mention backup planning for all databases and configuration items in the IT infrastructure of the regional center.

## 5. The Local Level of the Infrastructure

Here, we will briefly present the current state of the sensor level. The first sensor network is being built for open wheat blocks on the territory of the Institute of Plant Genetic Resources in the town of Sadovo. The experimental setup consists of multiple sensors used to monitor soil, leaf wetness, and air properties. The sensors that we are using are given in Table 1.

**Table 1.** Sensor summary and reporting values.

| Sensor | Environment | Measured Values |
|:---:|:---:|:---:|
| ADCON EnviroPro 40 | Soil | Four levels: moisture, salinity (electrical conductivity), and temperature |
| Seeed Studio 101990667 | Soil | Temperature, moisture, and electrical conductivity |
| ADCON WET Leaf Wetness | Leaves | Leaf wetness |
| Seeed Studio 314990738 | Leaves | Leaf |

The ADCON EnviroPro 40 sensor estimates soil moisture with a resolution of 0.01% and an accuracy of ±2%; salinity (or electrical conductivity) with a resolution of 0.001 dS/m and an accuracy of ±5%; and temperature with a resolution of 0.01 °C and an accuracy of ±1 °C. The sensors are located at four levels—at a depth of 0 cm (soil surface), 10 cm, 20 cm, and 30 cm. The device performs precise and repeatable measurements; therefore, in

our pilot setup, we use it as a reference device to validate other low-cost sensors such as the Seeed Studio 101990667. The Seeed Studio 101990667 sensor estimates soil temperature with an accuracy of $\pm 0.5$ °C and a resolution of 0.1 °C, a moisture accuracy of $\pm 2\%$, and a resolution of 0.03% (below 50%) and 1% (above 50%). The measured electrical conductivity is within the range of 0~10,000 μs/cm, with an accuracy of $\pm 3\%$ and a resolution of 10 μs/cm. The sensor provides built-in functionality for temperature compensation in the range of 0–50 °C. The experimental setup contains two more sensors for the estimation of the leaf wetness: the reference one is ADCON WET Leaf Wetness and the low-cost sensor is Seeed Studio 314990738.

The sensors are installed in accordance with their requirements on a pillar into a field of crops in Sadovo, a small town near Plovdiv. The pillar consists of a portable metal tube where electronics, power supply, and communications are placed in a box and the sensor groups are attached on the side, as shown in the photograph in Figure 4.

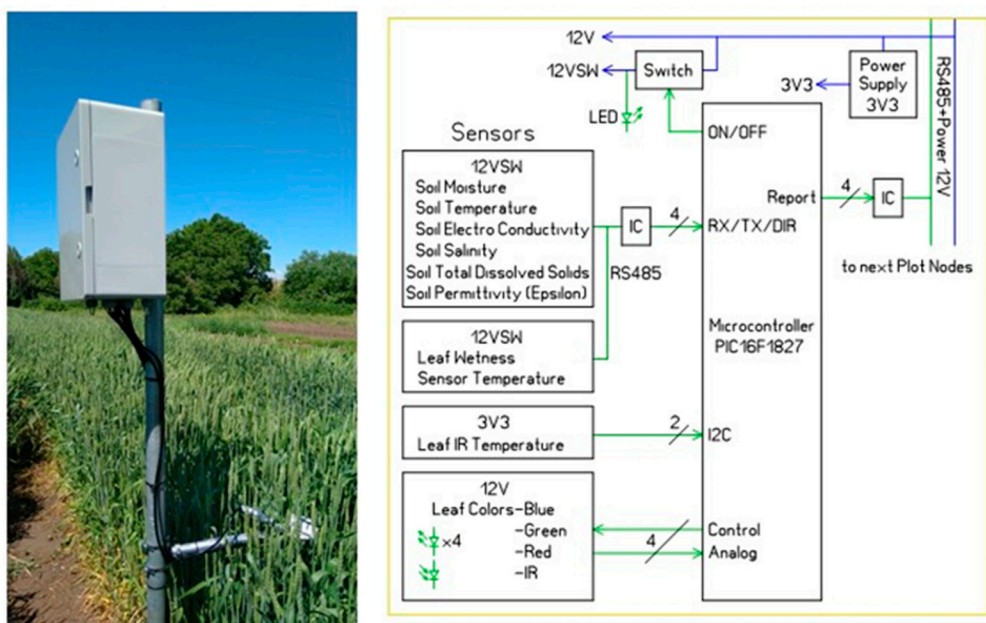

**Figure 4.** Experimental setup: picture (**left**) and flowchart (**right**).

The flow chart of the electronics is schematically presented on the right side of the figure. There are several items controlled by a PIC processor, such as a 12 volts for a soil group that includes two devices, 12 volts for a leaf wetness group of two devices, two more sensor groups reserved for infrared temperature measurements, and an RGB/IR sensor to measure the leaf color to estimate nitrogen content.

The measurements are performed in a special study field of 36 lots of wheat. For the pilot measurements, we chose lot number 17, as shown in Figure 5. Electricity is supplied via a cable to the pump room and the communication to the nearby building is wireless.

The current setup is operating until the harvest time and estimating the above-mentioned values every 30 min. The results in both raw data and graphics can be found on our data hub site, and they are publicly accessible [19,20].

The soil moisture measurements within the interval from the end of May to the beginning of June 2022 are shown in Figure 6. The color plots stand for the four soil depth levels: 0 cm or ground zero, 10 cm depth, 20 cm, and 30 cm. As expected, the highest moisture is observed in the top layer shortly after rainfall (20 and 31 May as well as 6 June). Later, due to the sun's heat and strong evaporation, the zero-level moisture went down in a few days. However, water penetration in the soil in time and depth is a complex function, described by the second Fick's law of diffusion [21]. Furthermore, this data can be used for better the understanding, modeling, and prediction of when and with what amount the crops must be watered, depending on the depth of their root system.

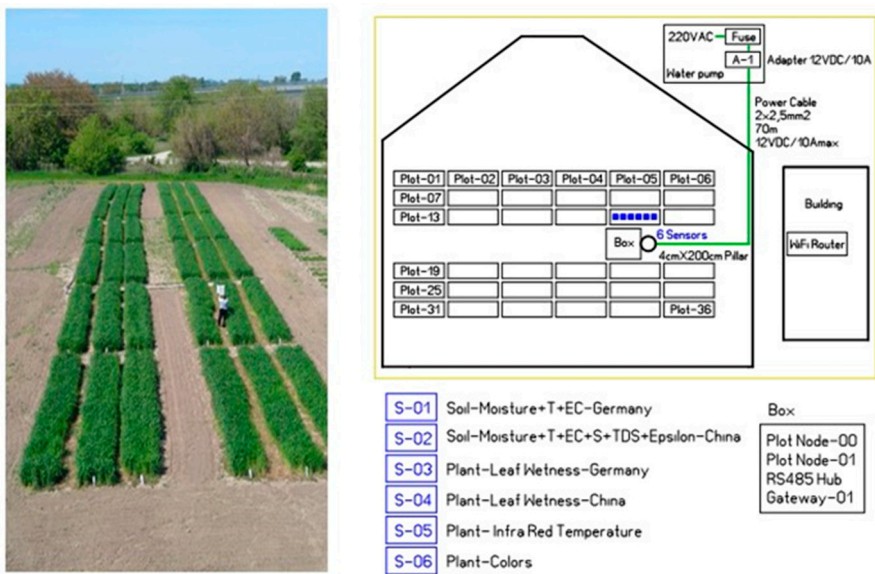

**Figure 5.** A picture of wheat fields and a schematic overview.

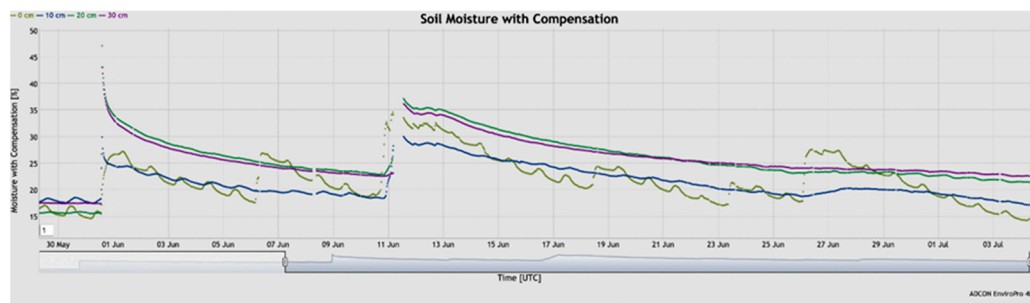

**Figure 6.** Soil moisture measurements in May and June 2022 in Sadovo, Bulgaria.

The next plot represents in-depth soil temperature measurements (Figure 7). The data were taken simultaneously, as in the previous graph, and both are physically connected. The rainfall events mentioned above can also be seen here as a temperature drop observed at the ground-zero level.

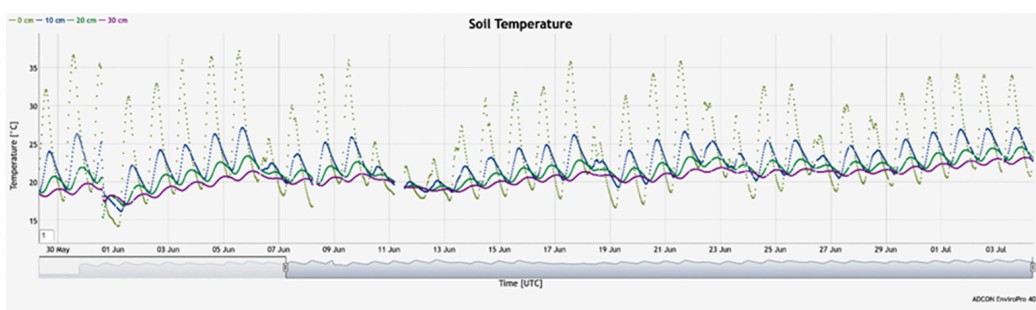

**Figure 7.** In-depth soil temperature measurements in May and June 2022 in Sadovo, Bulgaria.

With these preliminary results, we would like to validate our experimental setup and modeling assumptions and use them in production for the next season.

## 6. The Regional Data Center

The architecture of the regional data center includes two parts. The first one accomplishes communication to the local level to receive and store stream data arriving from the sensor networks. The second is a platform specializing in the delivery of various services

to farmers, using different approaches to model agricultural scenarios. One such service is, for example, the tracking, identification, and localization of anomalies in the vegetation of agricultural crops. In this section, the first part is presented in more detail.

The regional data center has a layered architecture (Figure 8). The first layer is made up of the physical sensor level presented in the article above (part 4—local level). In architecture, this layer is considered in the abstract as a combination of nodes and sensors. There are no restrictions on the number of nodes or the number and type of sensors that are included in them. The data generated by the sensors is accumulated by the nodes and transmitted at fixed intervals of time to the next layer—the adapters.

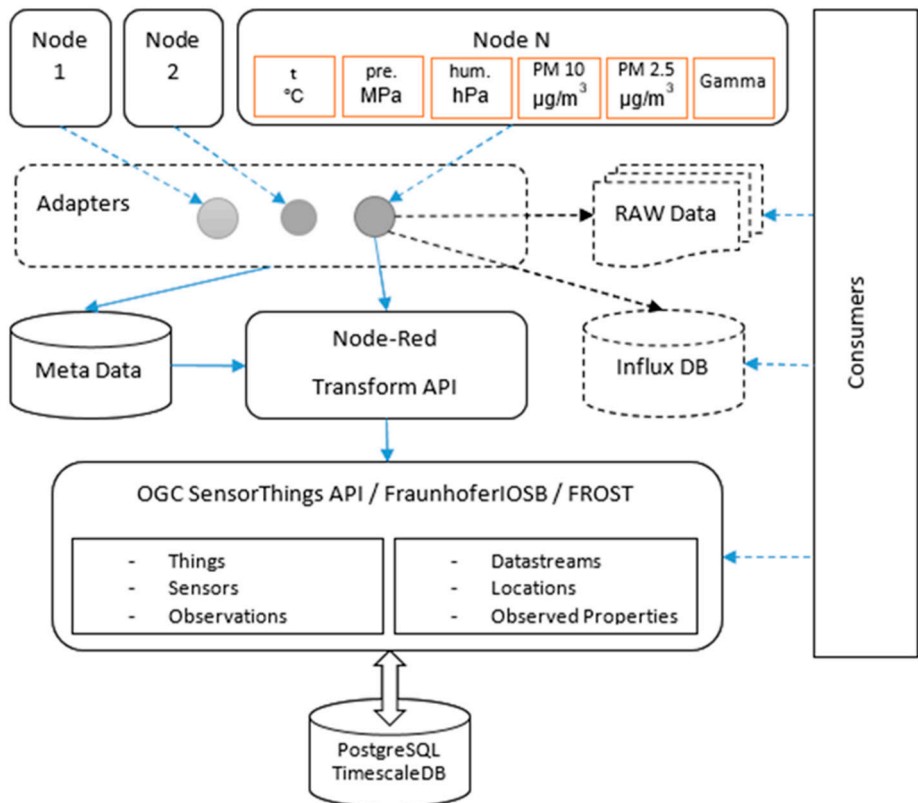

**Figure 8.** Architecture of the regional data center.

In order to minimize the data package, its structure is simplified as much as possible—a sequence of values separated by an interval. In addition to the data from the sensors, the package also includes information identifying the node itself, the exact time of transmission, and the type of sensors with which the respective measurements were made. An example data package looks like this:

G99 R775 N509 V 61 Z + 000 N 00 0017 0023 0972.7 + 02.9 066 0119 3.3 1638975076 31.13.201.118 16.037

Over time, the development of new components improves or changes due to alterations in the components and sensors used. Some components and sensors are discontinued, or their availability at suppliers is temporarily depleted, which requires their replacement with others that are not always close analogs. This inevitably leads to a change in the structure of the data packet. For this reason, the regional data center must be able to receive data from various types of nodes built with heterogeneous components and sensors. This problem is solved through the adapter layer. This layer contains a software component—an adapter, which receives the raw data packets from the nodes. A separate adapter is created for each type of node, or more precisely, for each type of data packet. Its purpose is to unify the raw data packet by eliminating the differences caused by the variation in the readings of the sensors used in the different nodes of the same entity.

Another role of the adapters is to correct the data from these nodes and sensors, which are known to give fixed deviations from the actual performance. They also identify and signal values that are outside the expected range. This helps in the early detection of defective nodes and components. The information about the expected limit values of the different types of sensors is maintained in the metadata database. It also stores the data needed to correct certain values from specific nodes.

As a security feature, the adapters register the raw data that they receive. Currently, this register is a different file rollover for each adapter. This register has several useful features:

It ensures that no data are lost in the event of a crash in the lower layers of the data center.

If necessary, it allows for analysis of the obtained data without superimposing the additional meaning of their transformation and adaptation.

It is possible to provide raw data directly to interested consumers.

In the future, the register will move to the use of a NoSQL database, which will ensure the storage of raw data and allow easier access to them.

The data unified by the adapters are then transmitted to the next layer, which is made up of two different systems with different purposes. One of them is the Influx Database, which directly stores unified data. This database is used by consumers that are interested in the current and average values over a period of time, e.g., visualization of changes in values of certain entities on a map or graphs (for example with Grafana).

The second system in this layer is designed to transform the unified data packet into a REST request, which is passed to the last layer that provides SensorThings API. The transformation is performed through Node-Red, a system that allows the visual construction of transforming logic by using nodes with predefined business logic. This makes it possible to reduce the effort associated with the implementation of this layer as well as facilitate maintenance when it is necessary to add new rules for new sensors and nodes.

The transforming logic also handles metadata that contain descriptions of specific nodes and sensors. From a unified data packet submitted by an adapter, the transform logic generates a set of REST requests to the SensorThings API, one for each entity. An example body of a REST request looks like this:

```
{
"result": 132,
"resultTime": "2021-12-08T14:52:32.000Z", "phenomenonTime": "2021-12-08
T14:52:32.000Z",
"Datastream": {
"@iot.id": "932"
}
}
```

The last layer is the implementation of the SensorThings API [22]. This is a standard of the Open Geospatial Consortium, which "provides an open, geospatial-enabled and unified way to interconnect the Internet of Things (IoT) devices, data, and applications over the Web." The FROST of the Fraunhofer Institut IOSB was selected for the implementation of this layer [23].

It provides a standard way to manage and retrieve observations and metadata from heterogeneous IoT sensor systems. The data are stored in a PostgreSQL database with a TimescaleDB extension. This significantly speeds up the recording and retrieval of data based on time intervals.

## 7. Data and Knowledge in the Regional Data Center

The main data repositories in the regional data center are databases and ontologies. The databases as mentioned earlier in the paper contain the dynamic data received from the various IoT nodes or from real-time data entered by the users of the data center. The

underlying knowledge for growing different crops is stored in ontologies. They are related to the main development phases of a crop, knowledge about the crop itself (family, genus, species, and their characteristics), knowledge about the soil in which the crop is grown, and knowledge about the activities that are performed under certain conditions to grow a crop. This basic knowledge forms the basis on which conclusions can be drawn about crop cultivation by comparing it with the data dynamically entered into the databases. Inferences can be made about the occurrence of certain events related to the growth of a plant or an entire block of a crop, or to determine the occurrence of various accidental events to which farmers should respond. In this part of the paper, the basic ontologies for storing the underlying knowledge about crop cultivation will be discussed and the idea of knowledge processing in the ZEMELA platform will be introduced, without presenting the platform in detail as this is not the purpose of the proposed paper.

*7.1. Ontologies in the Regional Data Center*

The ontologies in the regional data center are used to store the basic knowledge of crop cultivation. They can be divided into three types:

- Domain ontologies—these are basic knowledge that can be used for any crop cultivation. They include information generally accepted in the crop production domain, such as a plant taxonomy, soil types, etc.
- Event ontologies—these are basic knowledge related to the development of a specific crop type, i.e., the main phases of crop development or major events related to the cultivation of the specific crop. In addition, in these ontologies, exceptional events that may occur during the cultivation of a crop are described, such as drought, diseases, weeds, etc.
- Task ontologies—these are basic knowledge about the activities that need to be performed when an event occurs in the process of growing a specific agricultural crop.

These three types of ontologies will be discussed in more detail as part of the regional data center knowledge.

7.1.1. Domain Ontologies

Two domain ontologies have so far been developed in the regional data center. One ontology represents plant taxonomy and the second one, the soil types in Bulgaria. Both ontologies represent commonly accepted knowledge that is essential in the cultivation of different crops. On the one hand, the type of crop and its characteristics in terms of what species it is, genus, and family determine its characteristics; on the other hand, the type of soil is related both to the development of a particular type of crop and to the maintenance of soil moisture. The micronutrients contained in a particular soil type also indicate with what additional fertilizers it should be enriched in order to achieve the maximum yield from a given crop.

The ontology that contains the plant taxonomy according to [24] is called GenBankOntology. It represents the major families, subfamilies, genera, and species of plants (Figure 9).

For each Ordo, families, subfamilies, genera, species, subspecies, and varieties are presented. For families, subfamilies, and species, we have a description, which is represented in the ontology by annotation properties rdfs: comment, image, and synonyms. Rdfs:comment gives a short description of the family or species in English. In synonyms, known synonyms of the family or species are recorded, and in image, an image of the representative is given. In this way, information generally accepted for these families and species is stored. The description of species is not limited to the information in the annotation properties; a class descriptor has been added to the ontology that has Evaluation_characteristics as a subclass, which in turn provides additional information about the species according to the EURISCO [25] standard (Figure 10).

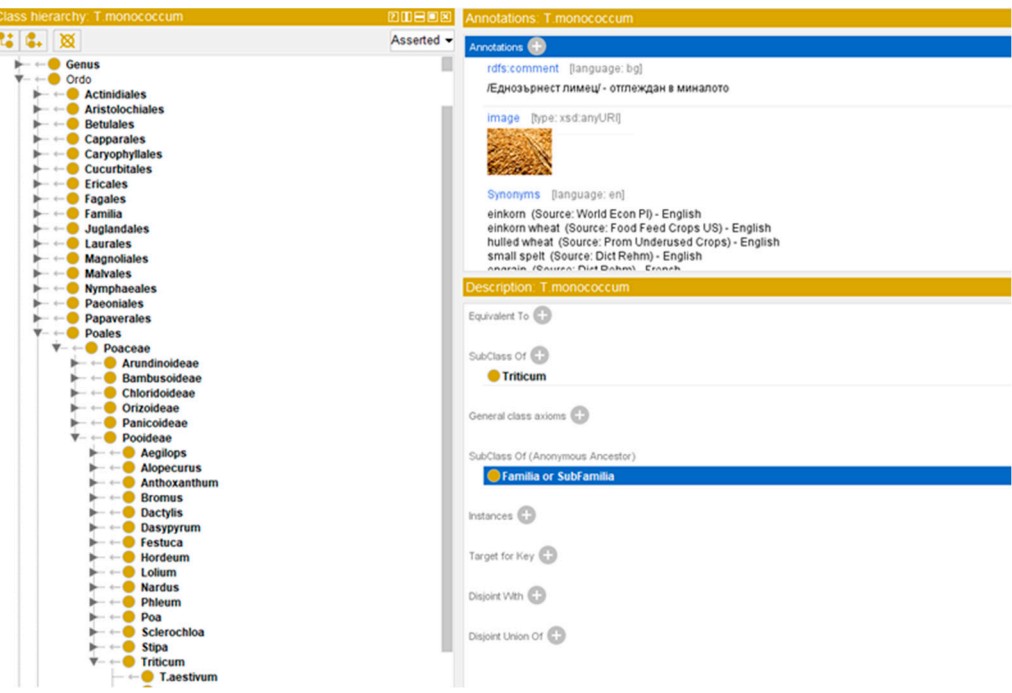

**Figure 9.** Plant taxonomy in the GenBankOntology.

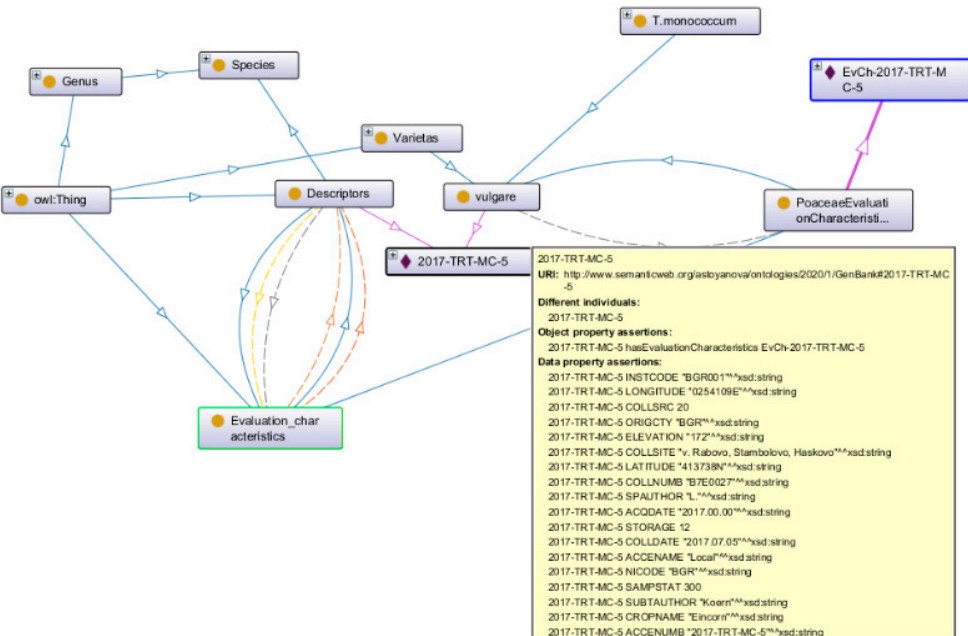

**Figure 10.** Descriptor and Evaluation_characteristics classes.

For example, the vulgare variety has Descriptor 2017-TRT-MC-5 and includes descriptive characteristics according to the EURISCO standard, such as where the species is found, who the author of its name is, where specimens of the species are kept, etc. This information about the species and the specimens of these species was taken from colleagues at the IPGR (Institute of Plant Genetic Resources), Sadovo, where the largest genebank in Bulgaria is located. It contains over 60,000 specimens of plant genetic resources.

All species or varieties have evaluation characteristics. They are presented by the Evaluation_characteristics class in the ontology and denote some specific characteristics for the species. For example, vulgare is a variety of the T.monococcum species and it has some

Poaceae evaluation characteristics. It is an EvCh-2017-TRT-MC-5 entity that is presented in Figure 11.

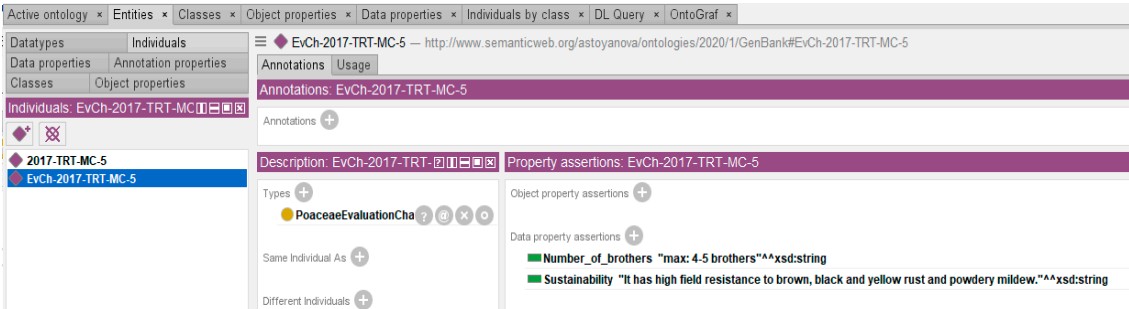

**Figure 11.** Evaluation characteristics of EvCh-2017-TRT-MC-5.

The evaluation characteristics are represented by the Data properties in the ontology. They make it possible to give specific values for the corresponding species' characteristics.

The second domain ontology that we have developed and is used in the regional data center is the ontology for soil types in Bulgaria—the SoilsOntology. This ontology represents the taxonomic vocabulary of soils in Bulgaria using the FAO global system [26] for the taxonomy (Figure 12). Soil type is extremely important in crop production as it influences activities such as crop watering and fertilization. For example, the effect of nitrogen fertilization also depends on the soil type and chemical elements—even if we add a lot of nitrogen into the soil, it will not be absorbed effectively if the pH values are low (in the case of acid soil), or if there is not enough carbon (organic matter) or trace elements. In addition, if the soil is sandy and there is heavy rainfall, nitrogen is easily washed away and the plants remain hungry. The ontology was developed for the specific soil types in Bulgaria.

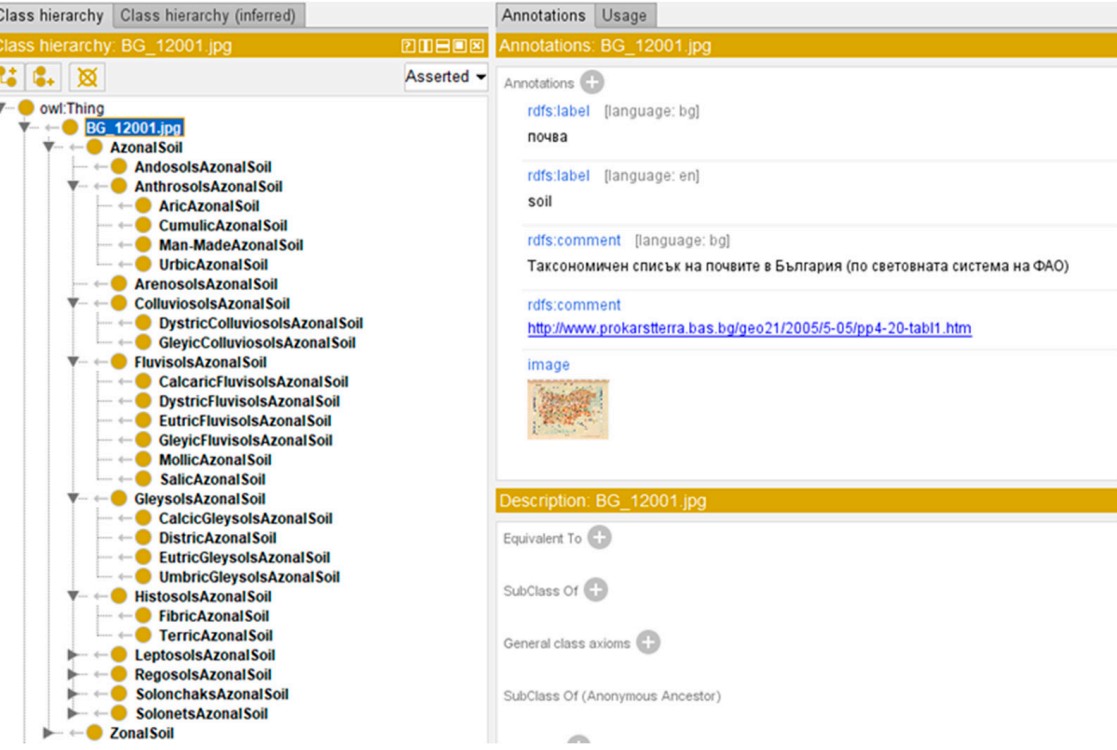

**Figure 12.** Bulgarian soil taxonomy in the SoilOntology.

In addition to the soil taxonomy, the ontology also presents the zonation of soils according to the crops that are suitable to be grown in these zones. A total of 16 wheat regions have been identified on the basis of the most suitable soil characteristics and the climatic conditions for wheat development. Each region has characteristics that are described as Data properties in the ontology. In *EighthWheatRegion* for example, the dominant soils are *LuvicChernozemsZonalSoil* or *LuvicPhaeozemsZonalSoil*. In addition, the data properties for this region show that the lowest percentage of humus is 1.5%; the highest percentage of humus is 2.4; and the suitability of the soils for the requirements of the crop (Bonit score—Benignity) is between 70 and 89, indicating that the soil is suitable for wheat cultivation.

7.1.2. Event Ontologies

Event ontologies were developed to represent the events that are important in growing different agricultural crops. An event can be defined as something that happens while a crop is grown (with the plant itself or with the environment). The events defined in crop cultivation are consistent with the event model presented in [27]. It was developed for the needs of the ZEMELA [1] intelligent farming environment. In terms of the ontologies proposed for use in the ZEMELA platform, we have divided the events into two main types—domain events and exceptional events. It can be said that from a practical point of view, this separation of events is quite sufficient. The two types of events can be defined as follows:

- domain event—it is related to the physiological development of the agricultural crop.
- an emergency event—it is related to the occurrence of conditions unfavorable to the cultivation of the agricultural crop and to deviations in the occurrence of domain events.

Thus, with these two definitions, we can say that we are modeling a template for the creation of event ontologies. The template will consist of a root, which we will call crop events, and two types of sub-events: domain and emergency, respectively. By developing these types of ontologies for different crops, it will be possible to track crop development through the ZEMELA platform. Event ontologies will represent the main knowledge base of plant development.

This paper briefly discusses the development of WheatEventOntology—an event ontology for winter wheat cultivation [5]. The ontology represents the two types of events for winter wheat cultivation (Figure 13). For the purposes of this paper, one of each of the two event types will be represented.

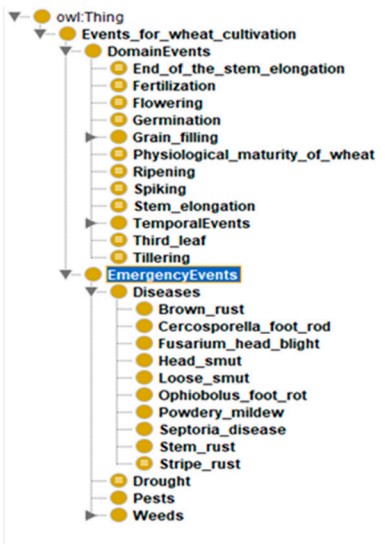

**Figure 13.** Taxonomy of event concepts.

Domain events are events related to the development of the plant itself ('phases of development'), e.g., germination, flowering, and pollination. Each of these phases is associated with the appearance of certain traits and the fulfillment of certain conditions to be considered as having occurred in a crop.

Figure 14 presents the stem elongation domain event. The condition for this event to be fulfilled for winter wheat is that the lowest internodes of the central stem appear 2–5 cm above the soil. This requirement is shown by the Equivalent To axiom. An object property "has_first_stem_node" is defined in the ontology for which we must have values between 2 and 5 cm.

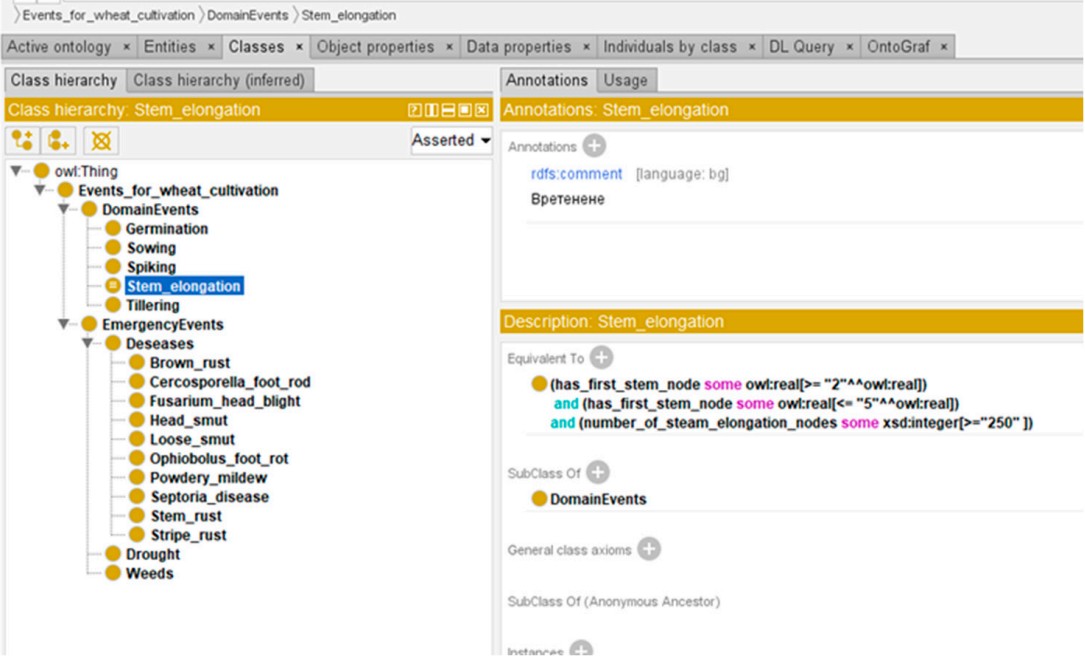

**Figure 14.** The Equivalent To axiom about the stem_elongation event.

If this condition is met and the number of nodes in this state is greater than 250 per square meter, then the crop may be considered to have entered the stem elongation phase or the stem elongation event has occurred.

Emergency events are defined in the winter wheat event ontology as variations in wheat development or the occurrence of wheat growing conditions that are unfavorable to wheat. Such events are disease, drought, and weeding. On the disease side, the ontology defines 10 diseases that are the most commonly occurring in winter wheat crops. Of course, these events can be updated with new diseases. In this paper, we will present one of the diseases for which we have a condition under which we can consider that the disease has occurred.

Figure 15 presents the condition for the occurrence of powdery mildew as an Equivalent To axiom. The stems and leaves of wheat are white and the stage of development is stem elongation or tillering. Additional events that are preventable can be defined here. Each of the diseases is characterized by temperature ranges in which it occurs, abiotic factors, and a combination of such which suggests the occurrence of a particular disease. Such preventive events could extend the ontology template with events in the future.

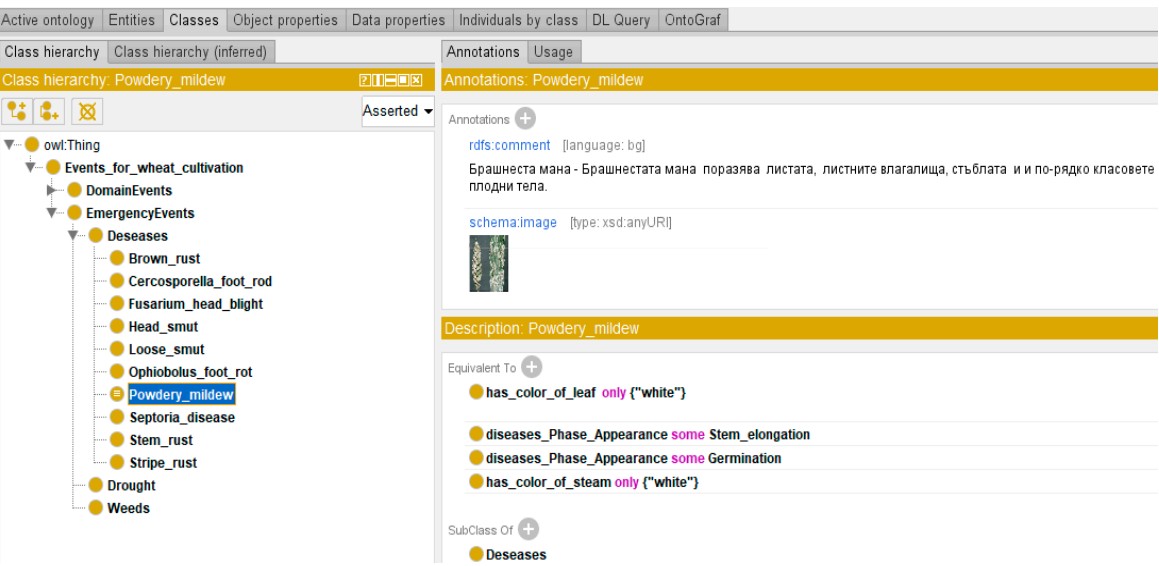

**Figure 15.** The Equivalent To axiom for the Powdery mildew disease.

## 8. Results

- The main results of the proposed study can be defined as follows:
- Development of the architecture of the regional data center in the Plovdiv region.
- Development of the data architecture.
- Development of the infrastructure of the regional data center at a local level and experimentation with different sensors.
- Presentation of data and knowledge in the regional data center in Plovdiv for smart agriculture.

Currently, the data from the different sensors at the local levels of the regional data center are collected in the proposed architecture of the regional data center in Plovdiv. The data can be viewed at https://meter.ac/gs/meteo/M93/history.html (accessed on 8 April 2023).

The architecture of the regional data center allows the raw data, which are generated by different types of sensors and grouped in different nodes with different architecture, to be unified and accessible to third systems through a standard interface defined by OGC. This enables the integration of the regional data center into independent workflows, analyzing data in real time as with the intelligent components in the ZEMELA platform.

The ontologies that have been developed are an important part of the knowledge stored in the regional data center. They are the background knowledge in the smart agriculture domain that is used by the intelligent components of ZEMELA. For example, WheatEventOntology was developed specifically for the purpose of growing winter wheat and is used by the intelligent components (personal assistants) of the ZEMELA platform to notify when domain events or emergency events occur in its cultivation.

Personal assistants in the ZEMELA platform were developed with the JaCaMo [28] technology for multi-agent system development and uses CArtAgo [29] as an implementation environment for artifacts that are used by agents in the system. This was the main reason for preferring to use ontologies to represent the underlying domain knowledge, and also the possibility to reuse them in other systems. For the moment, the first version of the personal assistant has been developed that uses ontologies as artifacts.

## 9. Conclusions

The main result for the proposed research is the implemented regional data center in the Plovdiv region. So far, there are three sensor networks developed at the local level from which data are collected: the first is at the Institute for Plant Genetic Resources located in the town of Sadovo (Plovdiv region); the second sensor network is deployed in the training

pasture of the Agricultural University of Plovdiv; and the third sensor network is placed in a tomato greenhouse located on the territory of the Maritsa Vegetable Crops Research Institute of Plovdiv. Up to now, the data are received and stored in the regional data center, but it should be considered that as the number of sensors increases, it will be necessary to change the architecture so that it is more scalable.

Another issue that is important to discuss is the high cost of sensors, which limits the requirement to cover larger areas with sensors and thus limits measurements. One possible solution we propose in the constructed infrastructure is that a small part of the sensors should be of high quality and accuracy and the rest can be cheaper, measuring with deviations, which can be compensated for by reference to high-accuracy sensors. This setup of sensors has been implemented in the Sadovo wheat fields—the biases of the less accurate sensors are taken into account when recording the data and this is managed by the metadata that are added to the raw data when they are recorded in the database.

The regional data center is an important component related to the development of smart agriculture systems. It maintains the main part of the data related to crop cultivation in the region. Based on the collected data, analyses and forecasts of the production and development of agricultural activities for the Plovdiv region can be easily made. The regional data center provides basic data for the operation of the ZEMELA platform, which includes intelligent components to support farmers. The smart components facilitate farmers to take decisions on crop cultivation and implement preventive actions to avoid various problems. The analyses and statistics derived from the data collected in the regional smart agriculture data center can also be useful for decision-making at the national level. This research will contribute to the more sustainable management of natural resources; it will reduce the environmental and climate impacts of agriculture as well as the use of pesticides and will improve the quality and safety of agricultural products, thereby ensuring food security and public health.

**Author Contributions:** Conceptualization, E.D., A.S.-D., A.T. and P.A; methodology, A.S.-D.; software, E.D., A.T., S.T., A.S.-D. and V.I.; validation, E.D., A.T., S.T., A.S.-D. and V.I.; resources, A.T. and A.S.-D.; data curation, A.T. and E.D.; writing—original draft preparation, A.S.-D., A.T., E.D. and P.A.; writing—review and editing, V.I. and A.S.-D.; visualization, A.T., E.D. and A.S.-D.; All authors have read and agreed to the published version of the manuscript.

**Funding:** This work is supported by the Bulgarian Ministry of Education and Science under the National Research Program "Smart crop production" approved by Decision of the Ministry Council №866/26.11.2020 and partially supported by the Scientific Research Fund under Grant № KP-06-X36/2, project BG PLANTNET "Establishment of National Information Network GenBank—Plant genetic resources".

**Data Availability Statement:** Publicly available datasets were analyzed in this study. This data can be found here: [https://meter.ac/gs/meteo/M93/history.html].

**Conflicts of Interest:** The authors declare no conflict of interest.

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
