# Peer review of "Architecture and Data Knowledge of the Regional Data Center for Intelligent Agriculture"

_information, doi:10.3390/info14040233_

Round 1

Reviewer 1 Report

This article presents the architecture of a Regional Data Center in the Plovdiv district. In line with the objective of the study, the authors tried to present as much information as they can. However, there are a lot of issues that need attention.

Abstract

Please improve the structure of the abstract-Follow the structure of the abstract
-Objective of the study
-The methodology used in the study
-The results
-The recommendations

Introduction

It would be really appreciated if you could provide citations for your work.
You have not included a single citation in the introduction. This is unacceptable

3. Methods

Due to the nature of your work, you should present the method clearly immediately after the Introduction. You should also try to present clearly the objective of the study.

Literature Review

Please you should have a clear subtopic stipulating the literature review with its own subtopics.

Results

The results of the study should also be presented clearly in your study. As it stands you did not include enough information in the results section.

Reviewer 2 Report

I have added some comments and questions to the paper. Please find them. 

Round 2

Reviewer 1 Report

You successfully addressed the majority of the concerns, but before the manuscript is published, double-check that you have included a discussion of the study's originality in the context of related work in the introduction.

Author Response

Dear Reviewer 1,
We would like to thank you once again for your time and your constructive feedback which helped us to improve the quality of our article.

Kind regards,
The authors

Reviewer 2 Report

Dear Authors, I have added some comments and questions to the text. Please find them. 

Author Response

Dear Reviewer 2,
We would like to thank you once again for your time and your constructive feedback which helped us to improve the quality of our article.

Kind regards,
The authors

Round 3

Reviewer 2 Report

It can be published

Author Response

Thank you very much again for taking the time to review our work and for your valuable and insightful comments! We are convinced that they will help us improve the quality of our article.

We made some changes to the references, which are noted in track changes in the manuscript.

Thank you again for taking the time and effort to review our research.